# Gut Microbiota Disruption in Hematologic Cancer Therapy: Molecular Insights and Implications for Treatment Efficacy

**DOI:** 10.3390/ijms251910255

**Published:** 2024-09-24

**Authors:** Patricia Guevara-Ramírez, Santiago Cadena-Ullauri, Elius Paz-Cruz, Viviana A. Ruiz-Pozo, Rafael Tamayo-Trujillo, Alejandro Cabrera-Andrade, Ana Karina Zambrano

**Affiliations:** 1Centro de Investigación Genética y Genómica, Facultad de Ciencias de la Salud Eugenio Espejo, Universidad UTE, Quito 170129, Ecuador; 2Escuela de Enfermería, Facultad de Ciencias de la Salud, Universidad de Las Américas, Quito 170124, Ecuador; 3Grupo de Bio-Quimioinformática, Universidad de Las Américas, Quito 170124, Ecuador

**Keywords:** leukemia, lymphoma, multiple myeloma, microbiota, cancer treatment

## Abstract

Hematologic malignancies (HMs), including leukemia, lymphoma, and multiple myeloma, involve the uncontrolled proliferation of abnormal blood cells, posing significant clinical challenges due to their heterogeneity and varied treatment responses. Despite recent advancements in therapies that have improved survival rates, particularly in chronic lymphocytic leukemia and acute lymphoblastic leukemia, treatments like chemotherapy and stem cell transplantation often disrupt gut microbiota, which can negatively impact treatment outcomes and increase infection risks. This review explores the complex, bidirectional interactions between gut microbiota and cancer treatments in patients with HMs. Gut microbiota can influence drug metabolism through mechanisms such as the production of enzymes like bacterial β-glucuronidases, which can alter drug efficacy and toxicity. Moreover, microbial metabolites like short-chain fatty acids can modulate the host immune response, enhancing treatment effectiveness. However, therapy often reduces the diversity of beneficial bacteria, such as *Bifidobacterium* and *Faecalibacterium*, while increasing pathogenic bacteria like *Enterococcus* and *Escherichia coli*. These findings highlight the critical need to preserve microbiota diversity during treatment. Future research should focus on personalized microbiome-based therapies, including probiotics, prebiotics, and fecal microbiota transplantation, to improve outcomes and quality of life for patients with hematologic malignancies.

## 1. Introduction

Hematologic malignancies (HMs), including leukemia, lymphoma, and multiple myeloma, are characterized by the uncontrolled growth of abnormal blood cells, affecting the blood, bone marrow, lymph nodes, and the immune system [1]. These malignancies pose significant clinical challenges due to their heterogeneity and variable treatment responses. In 2020, hematologic malignancies significantly contributed to the global cancer burden, with non-Hodgkin’s lymphomas (2.8% of global cancer incidence), leukemias (2.5%), multiple myeloma (0.9%), and Hodgkin’s lymphomas (0.4%) [2].

According to the Surveillance, Epidemiology, and End Results (SEER) data from the National Cancer Institute, the relative survival rate for hematologic cancer remains notably low among elderly individuals, particularly for acute myeloid leukemia (AML). For AML patients diagnosed between 2007 and 2013, the one-year survival rates were 18.2% for individuals aged 75–84 and 7.5% for those aged 85 and above [3]. However, recent therapeutic advancements, particularly for lymphoid malignancies, have improved survival rates. Notably, adults diagnosed with chronic lymphocytic leukemia (CLL) between 2007 and 2014 exhibited the highest one-year (91.8%) and five-year (76.5%) relative survival rates. Additionally, the five-year survival rate for acute lymphoblastic leukemia (ALL) increased from 31.6% in 1997–2002 to 39% in 2003–2008 [4,5].

Treatment strategies for hematologic malignancies depend on several factors, including the type of cancer, patient age, overall health, and spread and progression of the disease. Therapeutic approaches include stem cell transplantation, radiation, immunotherapy, targeted therapy, and chemotherapy [6]. Historically, chemotherapy has been the primary treatment employed to suppress cancer growth; however, combination therapies are now preferred because of their potential to reduce treatment duration and enhance efficacy. However, these regimens often result in adverse effects that affect physical well-being and overall quality of life [7,8]. Additionally, the emergence of drug resistance can compromise treatment efficacy, increasing the risk of relapse [9,10].

Chemotherapy and other treatments can also damage the intestinal mucosa [11] and disrupt the diversity of the gut microbiota, leading to dysbiosis [12]. The interaction between intestinal microorganisms and drugs is crucial in modulating drug activity and toxicity [13].

The human body harbors numerous bacteria, viruses, and fungi, predominantly residing in the intestine—the so-called gut microbiota. Alterations in this microbiota can impact an individual’s health and treatment response [14]. Several studies have demonstrated a reduction in the diversity of beneficial intestinal flora [15] and an increase in pathogenic bacteria following chemotherapy [16,17].

Researchers are exploring strategies to manipulate the composition and function of the microbiota to treat various diseases, including hematologic malignancies [18,19,20,21]. Moreover, characterizing the post-treatment microbiota can predict the risk of relapses, febrile conditions, and infections. Therefore, modulating the gut microbiota may enhance the efficacy of antitumor agents and reduce drug toxicity [22,23].

Recent interest has focused on bacteria-mediated cancer therapy due to its potential as a low-toxicity approach to cancer treatment [24]. Certain bacteria and their metabolites, such as *Salmonella*, have been shown to induced apoptosis in neuroblastoma cells without harming healthy tissues [25]. Similarly, bacteria like *Bifidobacterium* [26], *Clostridium* [22,27], and *Escherichia coli* [28] have shown promise in cancer treatment.

In this review, we conducted a literature search utilizing keywords such as microbiota, leukemia, lymphoma, and myeloma, combined with treatment approaches including chemotherapy, immunotherapy, and hematopoietic transplant. In the following sections, we describe the changes in gut microbiota after treatment with different drugs. Additionally, we compile and detail the main findings of research that focused on characterizing microbiota related to various treatments, specifically in patients with hematologic diseases. This review examines current therapies that alter the gut microbiota in patients with hematologic malignancies and explores the potential use of microbiota as an anticancer therapy.

### 1.1. The Link between the Microbiota and Cancer Therapy

The most common treatments for cancer include chemotherapy and radiation therapy, which use therapeutic drugs to inhibit the proliferation of cancer cells. Additionally, immunotherapy is increasingly employed, harnessing the immune system to target malignant neoplasms. Stem cell transplantation is often used in conjunction with these therapies to restore bone marrow cells that have been severely damaged by radiation or chemotherapy [29,30].

The intestinal microbiota plays a pivotal role in these therapeutic interventions. In adults, the microbiota includes mainly beneficial bacteria of the Firmicutes and Bacteroidetes phyla. In addition, phyla such as Actinobacteria, Proteobacteria, Fusobacteria, and Verrucomicrobia are found in smaller proportions [31]. These intestinal bacteria synthesize compounds, such as short-chain fatty acids, secondary bile acids, and other metabolites, that can increase or decrease the efficacy of therapeutic agents [32]. Conventionally, therapeutic drugs undergo metabolic transformations by hepatic enzymes in the liver, converting them into polar components for excretion. However, the intestinal microbiota also significantly contributes to drug metabolism, altering the chemical structure of drugs and modulating their absorption [33].

### 1.2. Drug Metabolism by the Intestinal Microbiota: Molecular Insight

The intestinal microbiome has a significant impact on drug metabolism through various molecular mechanisms (Figure 1). Several studies have identified the interaction between intestinal bacteria and drugs. For example, a study evaluated the ability of 76 intestinal bacteria to metabolize 271 drugs. The results revealed that *Bacteroides dorei* could metabolize 164 drugs, while *Clostridium* sp. metabolized 154 drugs [34]. This interaction can affect the drug bioavailability, toxicity, and therapeutic efficacy, contributing to interindividual variability in the response to treatments.

Routes of administration: Drug administration can occur through various routes, but regardless of the route, most drugs undergo metabolic transformations mediated by the intestinal microbiota. The most common routes are intravenous and oral administration [35]. Orally ingested drugs travel through the gastrointestinal tract, passing through the small and large intestines. In the intestines, these drugs are modified by both digestive enzymes and resident microorganisms. Once modified, drugs are absorbed by intestinal epithelial cells and transported to the liver via the portal vein. In the liver, they undergo further metabolic transformations by a variety of enzymes before entering the systemic circulation, where they are distributed throughout the body to affect various organs [36,37].

Additionally, some drugs are involved in an enterohepatic cycle. In this process, drugs that were previously metabolized in the liver can be excreted into the bile and returned to the intestine. During enterohepatic recycling, the intestinal bacteria secrete enzymes such as β-glucuronidase, β-glucosidase, demethylase, and desulfurase, which break down drug conjugates, allowing the reabsorption of active drugs [38].

In contrast, intravenously administered drugs enter the bloodstream directly and bypass the gastrointestinal tract. These drugs are distributed throughout the body via the circulation system, reaching various tissues and organs, including distal sites. However, some drugs or their metabolites may be secreted into the bile and return to the intestine, where the microbiota may intervene in their metabolism [37].

Drug–microbiota interactions: When the drugs come into contact with intestinal microorganisms, regardless of the route of administration, they may interact with these microbes directly through specific metabolic processes or indirectly by modulating the intestinal environment or host responses. Intestinal bacteria can metabolize drugs directly through various enzymatic reactions that differ from those performed by the human host [37,39], for instance, commensal bacteria, which is derived from the gusA gene [35]. β-glucuronidase plays a crucial role in metabolizing drugs that undergo glucuronidation in the liver. This process converts drugs into more water-soluble forms, facilitating their excretion through bile or urine [40,41]. However, when a glucuronidated drug reaches the intestines, β-glucuronidase can hydrolyze the conjugate, releasing the active form of the drug through deglucuronidation. The reactivated drug is reabsorbed into the bloodstream, resulting in enterohepatic recirculation. This process prolongs the presence of the drug in the body, affecting its pharmacokinetics, and may result in alterations in therapeutic efficacy or an increase in drug toxicity [42].

In addition to direct metabolic interactions, microbes can indirectly influence drug metabolism by modulating the host’s enzyme activity. For example, metabolites produced by the gut microbiota, such as short-chain fatty acids (SCFA), indoles, and secondary bile acids, can alter the expression and function of liver enzymes [43]. SCFAs such as acetate, propionate, and butyrate modulate hepatic cytochrome P450 (CYP) enzyme activity by activating peroxisome proliferator-activated receptor (PPAR), inhibiting histone deacetylases (HDAC) or altering the redox state. Also, indole derivatives generated by bacteria through tryptophan metabolism can inhibit CYP enzymes or induce conjugation reactions. Furthermore, gut bacteria interact with host transporter (TRP) proteins, such as P-glycoprotein (P-gp), influencing drug transport and metabolism indirectly [38].

Microbial metabolites can also compete with drugs for active sites on host enzymes, altering drug metabolism and efficacy. For example, p-cresol, a bacterial metabolite, competes with acetaminophen for host sulfotransferase enzymes, affecting its conjugation and elimination. This competition can alter both the efficacy and toxicity of the drug [31,44].

Importantly, these interactions are bidirectional: drugs can also alter the composition of the microbiota. For example, antibiotics can disrupt the structure of the microbiome, leading to long-term adverse effects on the immune and metabolic systems. In a study, mice treated with antibiotics (Abx) exhibited significant microbiota dysbiosis, which resulted in a fivefold reduction in B-cell lineages and a threefold reduction in T-cell lineages [45].

Therefore, drug–gut microbiota interactions, whether through direct metabolism or indirect effect on the gut environment, are critical in determining how therapies affect the body. Moreover, it is essential to highlight that this relationship is bidirectional: drugs can change the composition of the microbiome, and the microbiota can alter drug metabolism. Understanding this interaction is pivotal to increasing the efficacy of treatments but also to reducing the risk of long-term side effects.

Microbial enzymes and metabolic functions: Intestinal microbes metabolize drugs through various enzymatic reactions, including hydrolysis, hydroxylation, dihydroxylation, dealkylation, and reduction [46]. In addition, gut microbiota can remove functional groups via mechanisms such as N-oxide cleavage, proteolysis, and deconjugation. In contrast, host CYP enzymes primarily utilize oxidative and conjugative reactions. Microbial enzymes, such as hydrolases, lyases, oxidoreductases, and transferases, are widely secreted by intestinal microorganisms [47]. Hydrolases, including proteases, glycosidases, and sulfatases, break down large molecules into smaller products. Lyases break C-C and C-X bonds without the addition of water or oxidation. Oxidoreductases reduce functional groups such as alkenes, carboxylic acids, nitrous, nitrogen oxides, azo, and sulfoxides using cofactors such as NAD(P)H, flavin, and Fe-S clusters [36,48]. Transferases transfer functional groups between substrates using activated cofactors such as coenzyme A, ATP, or S-adenosylmethionine [37,49].

These enzymatic transformations affect the polarity, charge, and biological activity of drugs, influencing their shelf life and bioactivity in the human body. For example, the hydrolysis of glucuronides in the intestine can decrease the polarity of metabolites, allowing their reabsorption and extending their half-life. In addition, some microbial transformations can detoxify compounds, while others may generate toxic products [39,48]. Thus, microbial and host enzymatic activities create an interconnected metabolic network that modulates drug disposition and effects in the body.

Currently, a valuable tool called DrugBug assists in predicting specific enzymes and bacterial species involved in drug metabolism. This tool incorporates data from 491 human gut bacterial genomes and their 324,697 metabolic enzymes [50]. Understanding the interactions between the gut microbiota and drug metabolism is crucial for optimizing drug therapies and personalizing treatment strategies.

Drug bioactivity and microbiota influence: The activity of microbial enzymes significantly alters drug bioactivity, an area explored by the field of pharmacomicrobiomics. This discipline explores the drug–microbiome interrelationship, describing alterations in drug bioactivity. These changes include the activation of prodrugs, reduction of therapeutic efficacy through drug deactivation, reactivation via reabsorption of drugs from an inactive to an active form, and conversion of drugs into potentially toxic metabolites [33].

In drug activation, a prodrug is converted into an active form after metabolism. This metabolism is mainly regulated by the liver; however, enzymes produced by the intestinal microbiota also activate the conversion of the prodrug. For example, in treating hematological diseases like acute lymphoblastic leukemia (ALL), thiopurines such as 6-mercaptopurine (6-MP), 6-thioguanine (6-TG), and azathioprine (AZA-T) are commonly used [51,52]. These prodrugs are converted by different enzymes into active metabolites like 6-thioGTP and 6-thio-dGTP [51,52]. A study found that the role of the microbiota in thiopurine metabolism is essential, suggesting that the bacterial species capable of transforming these drugs are *Escherichia coli* (Proteobacteria), *Enterococcus faecalis* (Firmicutes), and *Bacteroides thetaiotaomicron* (Bacteroidetes) [53].

Another example is ixazomib citrate, a prodrug used in the treatment of multiple myeloma. Ixazomib undergoes activation through hydrolysis, deboronation, and N-dealkylation processes [54]. While ixazomib metabolism involves both cytochrome P450-dependent and -independent pathways, no studies have yet demonstrated a direct role for microbial enzymes in its activation. However, ixazomib acts as a proteasome inhibitor, targeting the nuclear factor kappa B (NF-κB) pathway in activated B cells [55,56]. Interestingly, short-chain fatty acids (SCFAs) produced by gut bacteria can suppress NF-κB activity and reduce inflammatory cytokine production. Although the microbiota’s direct role in ixazomib activation is unproven, this SCFA-related inhibition may enhance the efficacy of proteasome inhibitors like ixazomib [57].

In drug deactivation, the active form of the drug loses its therapeutic efficacy due to the action of intestinal microbial enzymes [58]. There is limited information on the deactivation of drugs used in the treatment of hematological diseases by bacterial action. However, some examples of deactivation are gemcitabine and cytarabine, which are widely used in the treatment of cancer. Specifically, cytarabine is used to treat acute non-lymphocytic leukemia, lymphocytic leukemia, and the blast phase of chronic myelocytic leukemia. One study analyzed the action of the enzyme cytidine deaminase against gemcitabine and cytarabine. They found that this enzyme produced by certain bacteria, such as *Mycoplasma hyorhinis*, inactivates these drugs, reducing their efficacy in cancer treatment [59].

In addition to drug deactivation, intestinal microbiota can also produce toxic intermediates during drug metabolism. Although such instances are rare, these toxic by-products can increase the risk of harmful side effects. For example, methotrexate (MTX), an antimetabolite commonly used to treat hematological diseases such as acute lymphoblastic leukemia, can be converted into toxic agents that exacerbate its gastrointestinal and hepatic side effects [60]. Metabolism of MTX occurs in the liver, generating 7-hydroxymethotrexate (7-OH-MTX), which is excreted into the bile and subsequently enters the gastrointestinal tract [61]. In the intestine, bacterial enzymes such as carboxypeptidase glutamate 2 (CPDG2) is responsible for cleaving the terminal glutamate residues of MTX and 7-OH-MTX, giving rise to metabolites such as 2,4-diamino-N(10-) methylpteroic acid (DAMPA) and 7-hydroxy-DAMPA. Additionally, *E. coli* expresses p-aminobenzoyl-glutamate hydrolase, which also contributes to this biotransformation. The low solubility of MTX and its by-products at acidic pH levels increases the risk of renal precipitation, leading to kidney toxicity. Furthermore, MTX can cause severe gastrointestinal toxicity, such as vomiting, diarrhea, and oral mucositis [62].

In drug reactivation, the enterohepatic circulation plays a crucial role. Initially, hepatic metabolism transforms the drug into conjugated metabolites, which are excreted in the bile and transported to the small intestine. In the small intestine, glucuronidase enzymes of the intestinal microbiota can deconjugate these metabolites, reactivating the drug in its active form [63,64]. The reactivated drug is reabsorbed and transported back to the liver through the portal vein and finally recirculated into the systemic circulation. This enterohepatic cycle prolongs the presence of the drug in the body, increasing its residence time and eliminating half-life [64].

Enterohepatic recirculation plays a crucial role in the pharmacokinetics of certain drugs, such as morphine [65] and methotrexate [66]. This process can lead to multiple peaks in drug concentration in the bloodstream. For example, methotrexate metabolism produces harmful by-products that can be reactivated through the enterohepatic pathway. After metabolizing in the liver, MXT by-products are excreted in the bile and pass into the small intestine. In the intestine, the CPDG2 enzyme breaks down these by-products, releasing the active form of methotrexate. The reactivated methotrexate is reabsorbed and recirculates through the enterohepatic system, returning to the liver and reentering the bloodstream [66,67]. This continuous cycle not only prolongs the presence of methotrexate in the body but also can enhance its therapeutic effect or its toxicity, depending on the accumulation and recirculation of its by-products.

In conclusion, the gut microbiota plays a pivotal role in modulating drug efficacy through a network of complex molecular mechanisms, in particular drug metabolism and bioactivity. Understanding these interactions is essential for the development of personalized medicine, allowing treatments to be tailored to the unique microbial composition of each individual. This personalized approach has the potential to improve therapeutic efficacy, reduce toxic side effects, and overcome resistance to conventional therapies.

### 1.3. Interactions between the Intestinal Microbiota and the Host Immune System

During drug metabolism, inflammatory effects can arise, often modulated by the immune system. Therefore, it is crucial to understand the interaction between the immune system, drug metabolism, and the intestinal microbiota to optimize therapeutic efficacy and minimize adverse effects. The interaction between the intestinal microbiota and host cells triggers a series of molecular processes. The main system involved in this interaction is the immune system. Epithelial cells and immune cells within the intestinal mucosa express pattern recognition receptors (PRRs), such as Toll-like receptors (TLRs) and NOD-like receptors (NLRs). Bacteria produce specific molecules, such as peptidoglycans and lipopolysaccharides (LPS), which are recognized by PRRs [68]. Upon detection of these molecules, PRRs activate and trigger an intracellular signaling cascade, activating the MyD88-dependent signaling pathway, which in turn activates NF-κB and MAPKs (ERK, JNK and p38) [69]. Activation of NF-κB and other signaling pathways results in the production of proinflammatory cytokines (TNF-α, IL-6, IL-12) and chemokines that recruit additional immune cells to the site of interaction [46,50,70].

Beneficial bacteria, such as *Bifidobacterium* and *Lactobacillus*, promote an anti-inflammatory environment and enhance antigen presentation. In addition, the microbiota can influence the differentiation of regulatory T (Treg) and Th17 cells. Under normal physiological conditions, the intestinal immune system experiences low-grade inflammation due to the constant exposure to microbial ligands that activate TLRs [49]. The commensal microbiota helps modulate this immune response by reducing NF-κB activation, thereby decreasing inflammation and promoting a balance between pro- and anti-inflammatory responses [50]. However, drug-induced mucosal damage can disrupt TLR signaling pathways, leading to inflammatory injury.

These interactions between the gut microbiota and the immune system are crucial in modulating the body’s response to cancer therapies. The microbiota not only influences inflammation and immune pathway activation but also directly impacts drug metabolism. This connection is essential to optimize the efficacy of cancer treatments and minimize side effects, highlighting the importance of considering the state of the microbiome in the development of personalized therapeutic strategies.

### 1.4. Chemotherapy

Chemotherapy drugs are essential in cancer treatment for disrupting the proliferation of malignant cells. These drugs are categorized based on their chemical composition and mechanisms of action [71]. Alkylating agents, such as cyclophosphamide (CP) and cisplatin, damage DNA to inhibit cell division. Antimetabolites, including cladribine, methotrexate, and 5-fluorouracil (5-FU), interfere with DNA and RNA synthesis, halting cell replication. Topoisomerase inhibitors, such as etoposide and topotecan, block enzymes essential for DNA replication, while mitotic inhibitors, like vincristine and paclitaxel, disrupt cell division. Antitumor antibiotics, such as doxorubicin and bleomycin, inhibit DNA transcription and replication, sometimes inducing DNA damage [72,73]. Additionally, corticosteroids are frequently used in combination therapies to manage side effects and occasionally exhibit anticancer properties. The choice of chemotherapy drugs or combinations depends on the specific cancer type and stage, with alkylating agents being among the most prescribed [71,72].

Chemotherapeutic agents exert their effects through multiple pathways, collectively described by Alexander and colleagues as the TIMER framework: Translocation, Immunomodulation, Metabolism, Enzymatic degradation, and Reduced diversity. This framework elucidates the multifaceted interactions between chemotherapy and gut microbiota, impacting treatment efficacy and patient outcomes [74,75].

Translocation and immunomodulation: Chemotherapy can cause bacterial translocation by damaging the gut epithelium, increasing infection risks, and affecting chemotherapy efficacy. For instance, a study by Viaud et al. (2014) investigated the impact of cyclophosphamide on the microbiota composition using mouse models. After 48 h of a non-myeloablative dose of CP, researchers observed the shortening of small intestinal villi, disruption of the epithelial barrier, interstitial edema, and focal inflammatory cell infiltrates [76]. Consequently, certain intestinal microorganisms like *Lactobacillus johnsonii*, *Lactobacillus murinus*, and *Enterococcus hirae* migrated to peripheral lymphoid organs, such as mesenteric lymph nodes and spleen. This bacterial translocation stimulates the activation of memory T helper 1 (Th1) cells and the conversion of naive CD4+ T cells to T helper 17 (Th17) cells, which secrete IL-17 and interferon-gamma (IFN-γ) [75,76]. These cytokines play a crucial role in mucosal healing and anticancer responses, underscoring the role of gut microbiota in influencing immune responses crucial for anticancer effects.

Metabolism and enzymatic degradation: Metabolic interactions between gut microbiota and chemotherapy drugs further highlight their impact on treatment efficacy and toxicity. The key enzymes involved in these interactions are bacterial β-glucuronidases (bβ-g). These enzymes, expressed by diverse gut bacteria, hydrolyze glucuronides to release active compounds, a process impacting the bioavailability of drugs. bβ-g is expressed by diverse bacterial phyla, such as Firmicutes, Bacteroidetes, Verrucomicrobia, and Proteobacteria [77,78]. For instance, in drugs like irinotecan (CPT-11) and diclofenac, bacterial β-glucuronidases remove glucuronide groups from drug metabolites, damaging the intestinal epithelium and intensifying gastrointestinal toxicity such as diarrhea [79]. Therefore, research indicates that inhibiting bβ-g activity can reduce drug toxicity in the gastrointestinal tract and enhance systemic drug efficacy, presenting potential therapeutic strategies to mitigate both local and systemic toxicity [78,80].

Reduce diversity: Chemotherapy often reduces microbial diversity, favoring pathogenic species while diminishing beneficial microbes. These changes exacerbate side effects like mucositis and diarrhea, altering immune responses. Lehouritis et al. (2015) demonstrated that the cytotoxicity of cladribine, etoposide phosphate, doxorubicin, and mitoxantrone was decreased by bacteria, consistent with other findings [81]. Zhou et al. (2017) found that methotrexate-treated mice showed significant changes in the gut microbiota. Particularly, Bacteroidales were the taxa that exhibited the most variation. Specifically, *Bacteroides fragilis* decreased, but an increase in macrophage density was also observed. This change in gut microbiota composition led to intestinal tissue injury [82]. Similarly, Nayak et al. (2021) reported that MTX affects the gut microbiota, leading to reduced host immune activation [83]. Iida et al. (2019) observed microbiota disruption in response to platinum chemotherapy, concluding that microbiota composition could influence optimal responses to cancer therapy [84].

Moreover, further studies on CP revealed significant shifts in microbial community structure in CP-treated mice. Mice were characterized by the presence of an increased proportion of Firmicutes/Bacteroidetes ratio and the absence of Verrucomicrobia, specifically *Akkermansia muciniphila*. CP treatment also led to increased abundance of bacteria in several classes, including *Bacilli*, *Clostridia*, *Coriobacteriia*, and *Mollicutes* [85]. These alterations suggest that CP promotes the proliferation of specific bacterial groups, potentially influencing immune responses and contributing to the observed dysbiosis.

Understanding these complex interactions through the TIMER framework provides valuable insights into optimizing cancer treatment by considering the significant role of gut microbiota. This highlights the potential for personalized therapeutic strategies that account for the microbiome’s influence on drug metabolism and patient outcomes [72,75].

### 1.5. Hematopoietic Stem Cell Transplantation

Hematopoietic stem cell transplantation (HSCT), commonly referred to as bone marrow transplantation, is a critical therapeutic strategy for patients with specific blood or bone marrow cancers. This procedure involves replacing dysfunctional or depleted bone marrow with healthy hematopoietic stem cells, promoting tumor cell destruction and generating functional cells to replace the damaged ones [86]. These replacement cells may originate from the patient’s own body or a compatible donor [87].

Clinical studies indicate that reduced microbial diversity during HSCT correlates with heightened proinflammatory immune responses and increased risks of complications such as graft-versus-host disease (GvHD) and mortality [88,89,90]. Several studies have shown that patients undergoing allogeneic HSCT (allo-HSCT) often experience a decline in microbiota diversity, characterized by an increase in pathogenic bacteria. A meta-analysis of 17 studies revealed that higher intestinal microbiota diversity significantly improves overall survival (HR = 0.66, 95% CI: 0.55–0.78), reduces transplant-related mortality (HR = 0.56, 95% CI: 0.41–0.76), and decreases the incidence of grades II–IV acute GvHD (HR = 0.41, 95% CI: 0.27–0.63). Additionally, higher abundances of Clostridiales were linked to better overall survival, while higher abundances of Enterococcus, γ-proteobacteria, and Candida were associated with poorer outcomes [91].

A study involving 1362 patients from four centers profiled the microbiota composition using 16S ribosomal RNA gene sequencing. Results showed consistent patterns of microbiota disruption across centers, with higher diversity at neutrophil engraftment associated with lower mortality. Notably, lower intestinal diversity before transplantation correlated with poor survival outcomes, suggesting early microbiota disruption [92].

Moreover, Kusakabe et al. (2019) observed a significant reduction in butyrate-producing bacterial genera and Bifidobacteria among allogeneic HSCT patients compared to healthy controls. Furthermore, they found that a decrease in microbial diversity predicts significantly poor post-HSCT survival [93].

Another study identified microbiota-based markers for monitoring complications in patients undergoing allo-HSCT. They discovered a link between an increase in Enterobacteriaceae and a higher risk of sepsis, as well as a decrease in Lachnospiraceae associated with a higher level of overall mortality [94].

Thus, the relationship between microbiota diversity and HSCT outcomes is evident, underscoring the importance of maintaining a diverse gut microbiome. Future research should focus on validating these findings and developing microbiota-based interventions to enhance patient outcomes in HSCT.

### 1.6. Immunotherapy

Promising immunotherapies include adoptive T-cell transfer, CpG-oligodeoxynucleotides (CpG-ODN), and immune checkpoint inhibitors (ICIs). Adoptive T-cell therapy relies on modifying T cells in an external environment before being reinfused into patients. CpG-oligodeoxynucleotides stimulate the immune system by mimicking bacterial DNA. Immune checkpoint inhibitors, which block proteins like CTLA-4, PD-1, and PD-L1, prevent tumor cells from evading the immune system and have been effective in advanced cancers [74,95]. These therapies aim to reactivate suppressed immune responses against cancer cells. Additionally, the gut’s immune cell interactions with commensal microbes may enhance immune responses, providing protection against pathogens [95].

CpG-ODN are synthetic DNA sequences that mimic bacterial DNA, designed to stimulate the immune system by activating Toll-like receptor 9 (TLR9). The efficacy of CpG-ODN in inducing antitumor responses is closely linked to the presence and composition of the gut microbiota. For instance, *Alistipes shahii* and Ruminococcus have been shown to enhance the effectiveness of CpG-ODN by promoting tumor necrosis factor (TNF) production, thereby amplifying CD8+ T-cell responses [96]. These commensal bacteria modulate the functions of immune cells in the tumor, which are essential for an optimal therapeutic response.

The therapeutic success of anti-CTLA-4 antibodies, such as ipilimumab, is partly dependent on the gut microbiota. Bacteroides fragilis and related species enhance the immune response through interleukin-12 (IL-12)-dependent Th1 signaling pathways [74,97]. In a mouse model, combining anti-CTLA-4 treatment with dextran sulfate sodium induced severe colitis. However, the administration of the probiotic Bifidobacterium mitigated colitis without compromising antitumor efficacy, likely through the action of regulatory T cells (Tregs). Conversely, certain Firmicutes species can exacerbate colitis, a common side effect of anti-CTLA-4 treatment [98].

The efficacy of anti-PD-1/PD-L1 therapies is also modulated by the gut microbiome. Bifidobacterium species, for example, enhance the effectiveness of anti-PD-L1 therapy by increasing CD8+ T-cell activation via dendritic cell pathways. Lactobacillus rhamnosus GG improves anti-PD-1 therapy outcomes through interferon-gamma (IFN-γ) and interferon-beta (IFN-β) pathways [74,99]. In germ-free or antibiotic-treated mice, the response to anti-PD-1 therapy was impaired. However, oral supplementation with *Akkermansia muciniphila* restored the efficacy of PD-1 blockade, suggesting a direct link between gut microbiota and immunotherapy response. For example, PD-1 and PD-L1 inhibit T cells, blocking the antitumor immune response. Moreover, studies are exploring the potential interaction between gut microbiota and PD-1 inhibitors [100]. Routy et al. (2018) found that the resistance to immune checkpoint inhibitors (ICIs) could be attributed to abnormal gut microbiome composition in patients with tumors. Metagenomic analysis of patients’ fecal samples revealed a correlation between clinical response to ICIs and the abundance of *Akkermansia muciniphila* [99].

There is still limited evidence of the interaction of gut microbiota with drugs used to treat hematologic cancer. Thus, more studies are needed to elucidate this relation.

## 2. Discussion

The availability of a wide range of hematologic cancer treatments has undoubtedly increased survival rates. However, concerns remain regarding toxicity, drug resistance, and side effects. Studies have shown that drugs frequently used in hematologic cancer treatment can alter the microbiota and reduce host immune activation. While evidence from hematologic cancer research and clinical trials is limited, several studies on other types of cancer suggest that the microbiota plays an important role in the cancer treatment response [101,102,103].

Figure 1 and Figure 2 and Table 1 and Table 2 in this review illustrate the significant imbalance in fecal microbiota due to various hematologic cancer treatments, emphasizing the need for strategies to maintain microbiota diversity to improve treatment outcomes. Using New Generation Sequencing (NGS), particularly 16S rRNA analysis, has allowed researchers to study the treatment’s impact on microbial profiles of cancer patients. Both autologous and allogeneic stem cell transplantation, as well as chemotherapy, have been linked to significant alterations in bacterial profiles, affecting patient outcomes and treatment effectiveness.

Chemotherapy primarily targets rapidly dividing cancer cells, but it also harms healthy cells, leading to adverse effects. Among these are effects on the gastrointestinal tract, where certain chemotherapy drugs induce apoptosis of healthy cells, causing mucosal damage (mucositis) and altering the gut microbiota [120]. Drugs such as 5-fluorouracil, irinotecan, methotrexate, capecitabine, cisplatin, oxaliplatin, and doxorubicin are known to cause side effects like mucositis, diarrhea, febrile processes, and pneumonia, which are often linked to changes in gut bacteria [106]. For example, Liu et al. (2021) in pediatric ALL patients reported that chemotherapy was associated with an altered microbiota, which triggered pneumonia [110].

Several studies have examined the impact of chemotherapy on the gut microbiota (Table 1). High-throughput sequencing techniques have demonstrated that chemotherapy disrupts the balance of microorganisms in the intestines. The studies included in this review identified significant changes in the abundance of bacteria from the Firmicutes, Bacteroidetes, Proteobacteria, Fusobacteria, and Actinobacteria phyla. Within the Firmicutes phylum, an increase in the family Lactobacillaceae, specifically the genus *Lactobacillus*, was observed. This increase may represent an adaptive response of the intestinal microbiota to restore balance following chemotherapy-induced damage. *Lactobacillus* species play a protective role by enhancing the intestinal barrier and boosting immune responses [121].

Additionally, the Enteroccaceae and Bacteroideceae families also showed increases, particularly in the *Enterococcus* and *Bacteroides* genera, respectively. This proliferation may result from reduced competition within the gut or immune system alterations caused by chemotherapy. *Bacteroides* help regulate immune responses and maintain intestinal homeostasis [122,123]. However, some *Enterococcus* species can be pathogenic, especially *Enterococcus faecalis* and *Enterococcus faecium*. These species can cause infections in immunocompromised patients [124].

In contrast, families like Lachnospiraceae, Bifidobacteriaceae, and Ruminococcaceae, which include genera such as *Blautia*, *Roseburia*, *Bifidobacterium*, and *Faecalibacterium*, showed a decrease in abundance. These genera are characterized by butyrate production except for *Blautia*, which produces acetate [125,126,127]. Acetate and butyrate are fatty acids that protect the intestinal mucosa and have anti-inflammatory properties. Therefore, depletion of these species could contribute to an inflammatory and damage-prone intestinal environment.

At the species level, *Lactobacillus fermentum* of the genus *Lactobacillus* was significantly increased following chemotherapy. *Lactobacillus* is a facultative anaerobic bacterium that enhances intestinal barrier defense by promoting mucus secretion [128]. Chemotherapy affects multiple components of the intestinal barrier, including the mucosal layer, epithelium, immune system, and vascular barrier [120]. Consequently, the increase in *Lactobacillus fermentum* may offer a protective role against intestinal damage caused by chemotherapy. A study evaluated the effect of *Lactobacillus*-fermented milk on 5-FU-induced mucositis in mice. The researchers found that the fermented milk prevented intestinal mucosal damage in mice by increasing villous preservation and reducing caliciform cells and inflammatory infiltration [129]. Smith and coworkers also concluded that a probiotic based on *L. fermentum* had the potential to reduce upper small intestinal inflammation caused by 5-fluorouracil [130]. Another study found that a combination of *L. fermentum*, *L. plantarum*, and vincristine protected mice from chemotherapy-induced damage [131]. This suggests that probiotics based on these species could help mitigate the side effects caused by chemotherapy.

Other species that increased in abundance after chemotherapy include *Coprococcus catus*, *Bacteroides coprocola*, *Escherichia sp.*, and *Enterococcus malodoratus*.

Species of the genus *Coprococcus* are obligate anaerobic bacteria that produce mainly butyrate. The presence of *Coprocococcus catus* and the production of butyrate may offer a protective effect against chemotherapy-induced inflammation because butyrate helps maintain the integrity of the intestinal barrier and modulates the immune response. The researchers found that butyrate could promote the efficacy of oxaliplatin by modulating the function of TCD8+ cells. The researchers observed that patients who responded favorably to oxaliplatin therapy had higher serum butyrate levels than non-responders, suggesting that butyrate may contribute to antitumor efficacy [132]. Therefore, the increase in butyrate-producing bacteria like *Coprococcus catus* could not only mitigate the inflammatory adverse effects of chemotherapy but also improve overall clinical outcomes in cancer treatment.

Another relevant observation is the increase in the abundance of species of the genus *Bacteroides*, especially *Bacteroides coprocola*. Studies suggest that *Bacteroides* plays a crucial role in reestablishing microbial communities post-chemotherapy. This genus has shown resistance to disturbances produced by both chemotherapy and antibiotic treatment. In a study of 117 individuals, they found that 21 bacterial species including *B. coprocola* exhibited a strong association with gut microbiome recovery after antibiotic therapy [133].

Mucosubstances such as glycosaminoglycans (GAGs) and host mucin are known to be responsible for the growth of the intestinal microbiota. GAGs and mucins are constantly supplied in the human intestine independent of nutrient intake by the host [134,135]. A study showed that *Bacteroides* degrade and assimilate GAGs and mucin, implying that this genus could survive even under malnutrition conditions [134]. In addition, their resistance to unfavorable environments may be due to their ability to penetrate the colonic mucus layer and reside within the crypt canals, a region that is more protected and less susceptible to stressors [136]. This suggest that *Bacteriodes* may aid in the recovery and repopulation of other beneficial intestinal species.

In addition, the increase in species such as *Escherichia* spp. and *Enterococcus malodoratus* can have negative implications. These bacteria have been associated with proinflammatory effects and toxin production, which can lead to gastrointestinal complications [137,138]. For example, *Enterococcus* infections are common in patients undergoing intensive chemotherapy causing prolonged neutropenia and mucositis. In a retrospective cohort study involving patients with acute leukemia, *Enterococcus* infections were a frequent complication, mainly associated with mucosal barrier damage in febrile neutropenic patients following chemotherapy [139]. Another study in children with lymphoblastic leukemia found that species such as *Enterococcus malodoratus*, *Ochrobactrum anthropi*, and *Actinomyces cardiffensis* were associated with chemotherapy-induced pneumonia [110].

The genus *Enterococcus* consists of Gram-positive bacteria known for their resistant and adaptability in adverse conditions. Some species, such as *Enterococcus faecalis* and *Enterococcus faecium*, have intrinsic resistance to common antibiotics [140]. One study found that 80% of *Enterococcus* isolates were resistant to vancomycin [141]. Therefore, the issue of infections or resistance related to *Enterococcus* after chemotherapy presents a significant challenge for healthcare systems. As a result, current efforts are focused on finding solutions to these problems. One potential solution is fecal microbiota transplantation (FMT), which has gained substantial interest from the scientific community in recent years. For instance, in a study, researchers used the feces of a 20-year-old donor to treat patients with vancomycin-resistant enterococci (VRE). The results indicated a significant decrease in *Enterococcus* abundance post-transplantation [142]. Likewise, a mouse model demonstrated that FMT in animals colonized with *Enterococcus faecium* VRE decreased infection by pathogenic microorganisms and restored the gut flora, including *Barnesiella* [143].

FMT has also been shown to reduce *Escherichia* species. In one study, mice that did not receive FMT experienced an increase in *Escherichia coli*, while those treated with FMT saw a significant reduction in the bacterium [144].

Furthermore, *Bifidobacterium* spp. and *Faecalibacterium prausnitzii* species showed decreased abundance. The loss of these beneficial bacteria can compromise the integrity of the intestinal barrier, increasing susceptibility to inflammation and infection. *F. prausnitzii* is the main butyrate-producing bacteria, and *Bifidobacterium* helps modulate the immune system [126,145]. Therefore, the reduction of these species can contribute to an inflammatory environment post-chemotherapy.

Bifidobacterium is a genus of Gram-positive bacteria that has been shown to inhibit tumor growth by activating the immune system and inducing apoptosis in tumor cells [145]. In a study with mice, they evaluated the effects of different probiotic mixtures containing *Lactobacillus* spp. and *Bifidobacterium* spp. The researchers found that probiotic mixtures could be potential therapeutic agents against intestinal mucositis caused by 5-fluorouracil [146].

Furthermore, *Faecalibacterium prausnitzii* is a commonly found bacteria in the gut microbiota that produces butyrate. Butyrate helps regulate inflammation and maintain the integrity of the intestinal barrier by modulating immune and epithelial cells. The gut microbiota also activates pattern recognition receptors (PRRs) such as TLRs, which modulate the immune response and can enhance the efficacy of treatments. For example, certain microbial metabolites like butyrate can promote Treg differentiation, modulating the inflammatory response.

Furthermore, a study by Wang and colleagues involving AML mice found that oral sodium butyrate (15 mg/kg) significantly reduced the leukemic burden in the bone marrow, peripheral blood, and spleen compared to control mice. The treated mice also exhibited reduced splenomegaly and improved intestinal barrier integrity. These results highlight the potential of butyrate as an adjuvant therapy in AML, particularly in combination with chemotherapy to reduce both disease progression and treatment-related complications [147].

On the other hand, another treatment that can drastically alter the intestinal microbiota is hematopoietic stem cell transplantation. The alteration caused by this therapeutic intervention can influence the success of transplantation and the occurrence of post-transplant complications such as graft-versus-host disease. Studies reviewed here indicate that HSCT affects the composition of the intestinal microbiota (Table 2).

In patients undergoing HSCT, there was a remarkable change at the phylum level, with an increase in Proteobacteria and a decrease in Firmicutes and Bacteroidetes. Proteobacteria include pathogenic bacteria such as *Klebsiella*, *Escherichia*, and Enterobacteriaceae, which dominate the gut microbiota after treatment. This dysbiosis is associated with adverse outcomes, including an increased risk of bloodstream infections (BSIs) and GvHD. Proteobacteria are Gram-negative bacteria are of clinical importance because of their high resistance to antibiotics [148]. A study of post-HSCT microbiota revealed that a higher abundance of Proteobacteria was associated with mortality, while patient survival was linked to higher levels of Lachnospiraceae [113]. Another study reported that the abundance of Proteobacteria after HSCT predicted bloodstream infections independently [149].

Additionally, it is crucial to restore the Firmicutes and Bacteroidetes phyla to prevent intestinal inflammation and reduce transplantation-related complications like infections, multiorgan failure, and GvHD. GvHD can exacerbate intestinal dysbiosis and cause morbidity and mortality after transplantation. A study analyzed 23 patients with steroid-refractory grade IV GvHD who received FMT. The results showed that Bacteroidetes and Firmicutes increased while Proteobacteria decreased after FMT, leading to an improvement in diarrhea and abdominal pain [150]. In another study, researchers used autologous FMT collected before allogeneic HSCT and tested it for intestinal pathogenic bacteria such as *C. difficile*. The results showed that autologous FMT intervention restored the microbiota after transplantation, with a reestablishment of Lachnospiraceae, Ruminococcaceae, and Bacteroidetes observed [151]. Another study also found elevated levels of Bacteroidetes following FMT [152].

At the family level, a decrease in beneficial bacterial families such as Lachnospiraceae and Ruminococcaceae was observed. These families are known to produce SCFAs such as butyrate. The reduction of these families can weaken the intestinal barrier, increase inflammation, and predispose patients to infections. These changes are prevalent in patients undergoing autologous SCT. Conversely, the Enterococcaceae family, which includes pathogenic opportunistic members, showed a significant increase.

Two studies evaluated the efficacy of FMT. The first study found that FMT in allogeneic HSCT recipients improved the alpha diversity of the microbiota, including Lachnospiraceae, Ruminococcoccaceae, and Bacteroidetes [151]. In the second study, patients who received FMT after neutrophil engraftment also showed recovery of alpha diversity and reduction of Enterococcus [153]. Another study conducted at two German tertiary centers demonstrated that FMT increased the abundance of Ruminococcaceae and Lachnospiraceae while suppressing Akkermansiaceae and Enterococcaceae [154].

Furthermore, the accumulated evidence from the cited studies indicates increases in the genera *Enterococcus* and *Streptococcus*. These genera include species resistant to many antibiotics and are often involved in hospital-acquired infections. It is concerning that these bacteria persist in the intestinal microbiota after transplantation because they can cause recurring infections and make treatment outcomes more complicated [155]. Different studies have linked *Enterococcus* and *Streptococcus* to relapses, indicating that these genera could be targeted for new diagnostic or therapeutic approaches to prevent relapse and improve overall survival after allo-HSCT [156,157].

An increase in the abundance of *Enterococcus faecalis* and *Enterococcus faecium* was also observed. These species are of particular concern because they are associated with antibiotic-resistant infections, especially in the context of GvHD. These species are known to persist in the intestinal microbiota despite the use of broad-spectrum antibiotics, which could contribute to persistent inflammation and further complications in patients with HSCT. The persistence of *Enterococcus faecalis* suggests a possible link between altered microbiota composition and post-transplant complications. A study suggested that *Enterococcus* may be used as a predictor of compromised microbiota and poor prognosis after allo-HSCT [158].

Additionally, beneficial species such as *Faecalibacterium prausnitzii*, which are associated with anti-inflammatory properties, showed a significant reduction. This decrease could have wide-ranging implications for intestinal health. Furthermore, antibiotics, which are frequently used prophylactically after a transplant, can increase the decline of these bacteria, especially during the transplant period. Hence, it is crucial to strengthen the presence of beneficial bacteria given that these species induce immune responses and anti-inflammatory effects. For instance, *Faecalibacterium* species stimulate the growth of regulatory T cells [159].

Data from HSCT studies suggest that microbial signatures could serve as biomarkers to predict patient outcomes, such as survival and the likelihood of developing GvHD. In addition, restoration of the gut microbiota through interventions such as FMT may be key to improving post-transplant outcomes in HSCT patients by favoring the recovery of beneficial bacteria.

On the other hand, immunotherapy can also alter the intestinal microbiota. In this review, we report only one study that characterized the microbiota in patients who underwent immunotherapy. Although information on this topic is limited, earlier sections discussed potential bacterial species that could enhance the efficacy of immunotherapeutic agents.

Finally, the limitations that we found during this review focused on several key aspects. First, many studies had relatively small sample sizes and lacked an adequate control group or stratification by risk factors. In addition, most studies focused on short-term changes in microbial diversity after treatment. Finally, several studies were limited to analyzing changes in the microbiota without linking them to clinical outcomes, making it difficult to establish clear causal relationships between microbiota disruption and health outcomes in hematologic patients. Future studies should address these limitations to make the findings more robust and clinically relevant.

In conclusion, the evidence presented highlights the complex interaction between hematological treatments and the gut microbiota. This highlights the critical need for strategies to maintain and restore microbial diversity. Changes in the microbiota caused by these therapies can compromise immune function, increase susceptibility to infections, and contribute to treatment-related side effects. Notably, there are significant alterations in bacterial populations, including an increase in pathogenic genera such as *Enterococcus* and *Escherichia*, as well as a decrease in beneficial species such as *Faecalibacterium prausnitzii*, *Bifidobacterium* spp., and Lachnospiraceae. Managing the microbiota may be crucial in improving clinical outcomes and reducing associated complications, such as antibiotic-resistant infections and graft-versus-host disease.

In this context, interventions like fecal microbiota transplantation could be promising. This approach aims not only to restore microbial diversity but also to reduce the adverse effects of treatment by strengthening the immune response and improving the integrity of the intestinal barrier. Integrating approaches that consider the microbiota as a determinant factor in therapeutic response could mark a significant advance in the treatment of hematologic cancer, opening new avenues for adjuvant therapies that maximize treatment efficacy and minimize post-treatment complications.

## 3. Future Directions

Hematologic microbiota research has led to new techniques for treating hematopoietic malignancies. For instance, Woerner et al. (2022) demonstrated dysbiosis in the circulating microbiome of patients with myeloid malignancies. This study determined that individuals with myeloid cancers have distinct dominant bacterial phyla and reduced alpha diversity compared to healthy controls [160]. Furthermore, according to Luthuli et al. (2023), many pathogenic organisms play an important role in the development of lymphomas. Their review study indicates that *H. pylori* is associated with mucosa-associated lymphoid tissue lymphoma (MALToma), and antibiotic therapy for this bacterium has resulted in MALToma remission [161].

Moreover, Chen et al. (2024) employed a bidirectional Mendelian randomization approach to investigate the causal relationship between gut microbiota and hematologic malignancies. The study identified significant associations between the abundance of specific bacterial taxa and the risk of lymphoid and myeloid leukemia, Hodgkin lymphoma, malignant plasma cell tumors, diffuse large B-cell lymphoma, mature T/NK cell lymphomas, and myeloproliferative neoplasms [162].

Thus, studies on gut microbiota have demonstrated their potential to identify new biomarkers for hematological cancers. Future research could explore their application in non-invasive diagnostic methods.

In addition, future investigations should also focus on understanding the mechanisms underlying microbiota changes following cancer therapy and developing innovative therapeutic strategies to restore microbial balance. Targeted treatments, such as probiotics, prebiotics, and fecal microbiota transplantation (FMT), show promise in reducing treatment-related complications and improving outcomes for patients with hematologic cancers.

Prebiotics, including oligosaccharides, are food components that stimulate the growth of beneficial microorganisms within the intestinal microbiota [163]. Similarly, probiotics are beneficial microorganisms that restore the intestinal microbiota and prevent the growth of pathogenic bacteria. Both probiotics and prebiotics could induce and promote physicochemical conditions unfavorable to cancer cell survival [164]. Additionally, they may contribute as complementary therapies to enhance the efficacy of treatments such as chemotherapy while reducing associated adverse effects [165].

Similarly, FMT is particularly effective in restoring microbiota homeostasis, which may improve cancer patients’ responses to treatment [166,167]. For instance, the chemotherapeutic agent cyclophosphamide, used for lymphoma, leukemia, and other types of cancer, has adverse effects such as alteration of the intestinal mucosa and recruitment of inflammatory cells. These adverse effects can alter the intestinal microbiota of the Lactobacilli and Enterococci genera [168,169]. However, FMT could deliver specific bacterial genera of the intestinal microbiota to reduce side effects and improve treatment efficacy [170]. Davar et al. (2021) demonstrated that FMT significantly enhanced therapy effectiveness in cancer patients who were previously non-responders, emphasizing the microbiota’s role in treatment response [171].

In this context, multiple clinical trials are investigating the association between gut microbiota composition and hematologic cancer. The NCT01600781 clinical trial, now completed, hypothesized that fiber supplementation could improve gut microbiota and enhance the immune response in patients with acute lymphocytic leukemia [124]. Similarly, the NCT05045443 clinical trial evaluated the biological effect of curcumin on the microbiota in patients with acute lymphoblastic leukemia. Moreover, the NCT02928523 trial analyzed the use of autologous fecal microbiota transplantation in patients diagnosed with acute myeloid leukemia treated with intensive chemotherapy and antibiotics to restore their intestinal microbiome. The NCT03678493 trial focused on fecal microbiota transplantation in patients with acute myeloid leukemia undergoing chemotherapy and Allo-HCT. Furthermore, the NCT04935684 trial aimed to assess fecal microbiota transplantation in allogeneic hematopoietic stem cell transplantation to prevent complications and improve patient outcomes [172].

Additionally, NCT03316456 is a clinical trial that aims to identify the association between microbiota and chemotherapy-induced gut barrier damage in acute leukemia patients. NCT03797170 investigates the role of gut microbiota in mediating immune activation in response to chemotherapy in patients diagnosed with large B-cell lymphoma [172]. However, more studies are needed to further understand the association between gut microbiota composition and the response to chemotherapy treatment in patients with hematologic cancers.

Despite the effectiveness of modern treatments for hematologic diseases, a significant proportion of patients relapse, highlighting the need for novel therapeutic approaches [173,174]. Limited studies and clinical trials have explored the potential use of microorganisms as anticancer agents, primarily using animal models. Dang et al. (2001) investigated the use of anaerobic bacteria to kill tumor cells in poorly vascularized areas. They observed that mice died 16 to 18 h after the bacteria proliferated in hypoxic tumor regions, possibly due to the release of lethal toxins by the bacteria. Consequently, they eliminated the toxin gene in the *Clostridium novyi* strain and found that the modified strains were less toxic and effectively killed tumor cells. Their anticancer potential increased when combined with chemotherapy, a method known as bacteriolytic therapy [21].

Modified or attenuated bacteria are increasingly recognized for their low toxicity towards normal cells, making them important candidates for cancer treatment. A study in an animal model of lymphoma showed that an attenuated strain of *Salmonella Typhimurium* (an avirulent mutant) activates the immune system by stimulating the activation of CD8+ and natural killer T cells, resulting in the development of an effective systemic and local antitumor response [173].

Furthermore, another investigation evaluated the efficacy of Salmonella immunotherapy in mice bearing B-cell non-Hodgkin’s lymphoma (B-NHL) cells undergoing chemotherapy. The study revealed that Salmonella significantly delayed tumor growth and prolonged the survival of the lymphoma animals [175].

Undoubtedly, the treatment of hematologic cancer has generated a growing interest in the modulation of the intestinal microbiota through probiotics, prebiotics, and fecal microbiota transplantation. However, these therapies present significant limitations and challenges that must be carefully evaluated in their clinical application. While probiotics, prebiotics, and FMT can offer important benefits, especially after oncologic treatment, their use also carries potential risks and side effects. The main limitation is the lack of solid evidence to support their use reliably and effectively. In addition, individual response to these therapies is highly variable; for example, prebiotics rich in carbohydrates can cause bloating and abdominal discomfort in some patients [176,177]. Moreover, dysbiosis can also be improved through diet. Recent studies suggest that the ketogenic diet may modify the diversity and composition of the gut microbiota, which in turn could influence the efficacy of treatment for conditions such as cancer, epilepsy, and obesity [178,179]. Another study on acute leukemia suggests that a diet rich in vegetables and fruits modulates microbiota and immune responses [180]. They also suggest that exposure to plant diversity rich in glucan and related microbial communities promotes immune cell maturation and is linked to a lower incidence of childhood acute lymphoblastic leukemia [181].

In the case of FMT, there are risks of serious infections, exaggerated immune reactions, and transmission of unwanted pathogens, especially in immunosuppressed patients such as those who have received chemotherapy [182,183]. In addition, probiotics can induce systemic infections, gastrointestinal side effects, skin reactions, transfer of antibiotic resistance genes, deleterious effects of their metabolites, and abnormal stimulation of the immune system [176,184].

In summary, modulation of the gut microbiota by probiotics, prebiotics, and fecal microbiota transplantation (FMT) offers a promising approach in the treatment of hematological cancers but also presents significant challenges. Therefore, further research is needed to elucidate the mechanisms through which these strategies contribute to the efficacy of therapy.

## 4. Conclusions

This comprehensive review highlights the profound impact of cancer treatments, especially chemotherapy and hematopoietic stem cell transplantation, on the gut microbiota in patients with hematologic malignancies. Our key findings reveal that these treatments can disrupt the balance of microbial diversity, promote harmful bacteria, and lead to adverse effects like increased infection risk and weakened immune function. Comprehending these interactions is essential for improving cancer treatment outcomes.

The potential clinical applications of these findings are substantial. Preserving microbiota diversity through targeted interventions such as probiotics, prebiotics, and fecal microbiota transplantation could mitigate treatment-related complications and enhance therapeutic efficacy. Integrating microbiota-targeted therapies into standard treatment protocols for hematologic malignancies offers a promising avenue to improve patient outcomes and advance precision oncology.

Future research should focus on elucidating the mechanisms underlying microbiota alterations during cancer treatment, identifying biomarkers for predicting treatment responses, and developing personalized microbiome-based therapeutic strategies. Such efforts will be pivotal in harnessing the full potential of the gut microbiome to enhance the effectiveness of cancer treatments and improve the quality of life for patients with hematologic malignancies.

## Figures and Tables

**Figure 1 ijms-25-10255-f001:**
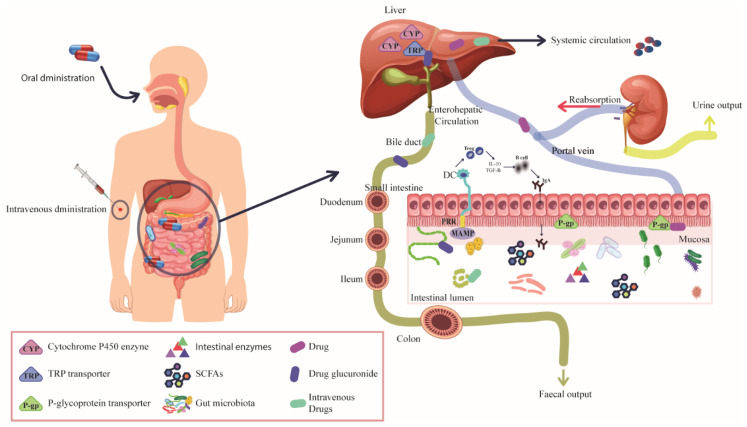
Mechanisms of interaction between intestinal microbiota, drug metabolism, and host immune response. Oral drugs interact with the microbiota, CYP enzymes, and TRP transporters in the gastrointestinal tract and liver, modulating their metabolism before entering the circulation. Intravenous drugs initially avoid these steps but may be modified by the microbiota and intestinal enzymes after excretion in the bile. Enterohepatic recirculation allows drugs metabolized in the liver to be reactivated by the intestinal microbiota, prolonging their presence in the body and affecting their efficacy or toxicity. The microbiota also modulates immune response and drug metabolism through the production of metabolites and influences enzymes.

**Figure 2 ijms-25-10255-f002:**
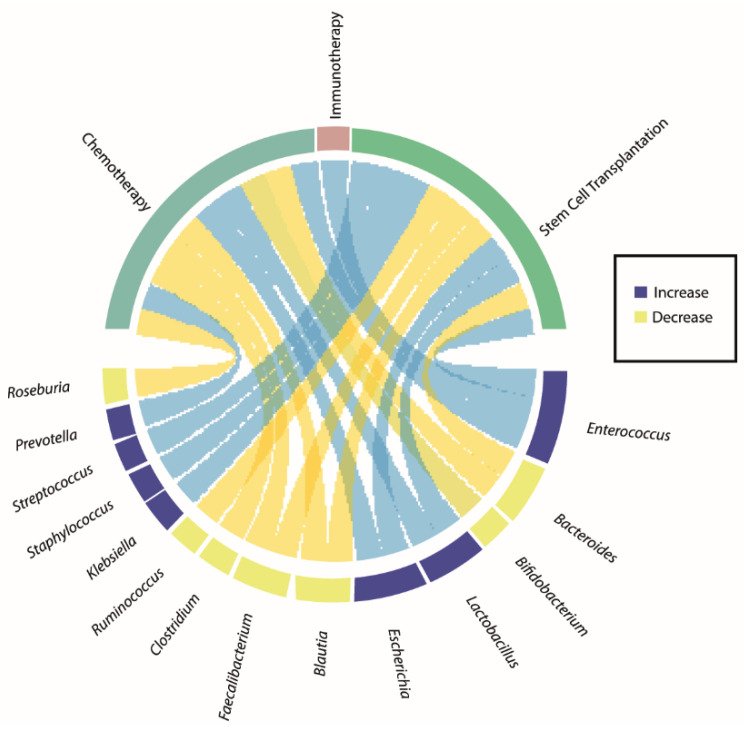
Role of the gut microbiota in hematologic cancer development and treatment response. A chord plot illustrating the relationships between microbial species and their response to cancer therapy (chemotherapy, immunotherapy, and stem cell transplantation). Immunotherapy is shown in pink, stem cell therapy in light green, and chemotherapy in dark green. The purple nodes represent increases, while the yellow nodes represent decreases. Blue ribbons represent the microbial species that increase in response to cancer therapy, while yellow ribbons represent species that decrease.

**Table 1 ijms-25-10255-t001:** Impact of chemotherapy on microbiota diversity in hematologic cancers, including multiple myeloma (MM), leukemia (LK), lymphoma (LP), acute myeloid leukemia (AML), non-Hodgkin’s lymphoma (NHL), acute lymphoblastic leukemia (ALL), allogeneic stem cell transplantation (allo-SCT), and autologous stem cell transplantation (auto-STC).

Chemotherapy							
Individuals	Hematologic Cancer Type	Bacteria		Abundance	Antimicrobial Administration	Outcome	Bibliography
34 patients	AML	Genus	* Lactobacillus *	Increase	Antimicrobial administration: fluoroquinolone, cephalosporin, azoles, echinocandins, amphotericin B, carbapenem, piperacillin/tazobactam, cefepime.	They suggest that microbiome measurements could assist with mitigation of infectious complications of AML therapy.	[104]
			* Blautia *	Decrease			
			* Prevotella *				
			* Leptotrichia *				
20 patients 5 controls	AML ALL	Species	* Lactobacillus fermentum *	Increase	Antimicrobial prophylaxis is not used.	The microbiota is altered by chemotherapy.	[105]
			* Coprococcus catus *				
			* Bacteroides coprocola *				
8 patients	NHL	Genus	* Bacteroides *	Increase	These patients received oracillin and cotrimoxazole as antibiotic prophylaxis.	They observed reduction in the abundance of organisms with anti-inflammatory properties and dysbiosis characterized by a significant establishment of Escherichia.	[106]
			* Escherichia *				
			* Blautia *	Decrease			
			* Faecalibacterium *				
			* Roseburia *				
			* Bifidobacterium *				
17 patients 17 controls	LK LP MM	Genus	* Bifidobacterium *	Decrease	Antibiotic: levofloxacin, cotrimoxazole, piperacillin, and tazobactam.	All bacteria and *Clostridium cluster XIVa* did not differ significantly in patients with or without antibiotics.	[23]
			* Bacteroides *	Increase			
		Species	* Clostridium cluster IV *	Increase			
			* Clostridium cluster XIVa *	Decrease			
13 patients		Species	*Bacteroides* spp.	Increase		Chemotherapy did not affect the intestinal microflora to any great extent.	[107]
29 patients	AML	Phylum	Proteobacteria	Increase	No antibiotics were used.	Chemotherapy significantly decreases the microbial richness of the gut microbiome.	[108]
			Firmicutes				
9 patients 11 controls	AML	Genus	* Enterococcus *	Increase	All patients received antibiotic prophylaxis.	The number of anaerobic bacteria decreases during treatment for AML, thereby increasing the proportion of aerobic and potentially enteropathogenic bacteria, such as enterococci.	[109]
			* Streptococcus *	Decrease			
		Species	* Bacteroides * species				
			* Clostridium cluster XIVa *				
			* Faecalibacterium prausnitzii *				
			* Bifidobacterium * species				
14 patients 44 controls	ALL	Species	* Enterococcus malodoratus *	Increase	Antibiotics were not used before taking stool samples.	They found that altered gut microbiota was associated with chemotherapy-induced pneumonia in pediatric ALL patients.	[110]
			* Ochrobactrum anthropi *				
			* Actinomyces cardiffensis *				
			* Bacillus altitudinis *	Decrease			
			* Afipia birgiae *				
			* Bifidobacterium tsurumiense *				
7 patients 7 controls	ALL	Genus	*Atopobium*	Decrease	During treatment, patients received trimethoprim-sulfamethoxazole for *Pneumocystis jiroveci* prophylaxis.	They suggest that the gut microbiota profile in children with ALL is different among healthy controls, and after chemotherapy is even different at the time of leukemia detection.	[111]
			*Bacteroides*				
			*Fusobacterium*				
			*Prevotella*				
			*Bifidobacterium*	Increase			
34 patients	AML	Genus	* Lactobacillus *	Increase	Received antibacterial (fluoroquinolone), antifungals (azoles), and antibiotics (carbapenem) after.	They report that low bacterial diversity in stool samples is associated with the development of infections during chemotherapy.	[104]
			* Blautia *	Decrease			
			* Prevotella *				
			* Leptotrichia *				

**Table 2 ijms-25-10255-t002:** Impact of hematopoietic stem cell transplantation and immunotherapy on microbiota diversity in hematologic cancers, including multiple myeloma (MM), leukemia (LK), lymphoma (LP), acute myeloid leukemia (AML), non-Hodgkin’s lymphoma (NHL), acute lymphoblastic leukemia (ALL), allogeneic stem cell transplantation (allo-SCT), and autologous stem cell transplantation (auto-STC).

Hematopoietic Stem Cell Transplantation								
Treatment	Individuals	Hematologic Cancer Type	Bacteria		Abundance	Antimicrobial Administration	Outcome	Bibliography
Allogeneic stem cell transplantation	94 patients	LK (44) LP (26) MM (6) Others (18)	Phylum	Proteobacteria	Increase	Antibiotics administered before and during treatment: vancomycin, fluoroquinolone, metronidaz, and beta-lactamd.	During allo-SCT, the diversity and stability of the intestinal flora were disrupted, which was associated with an increased risk of bloodstream infection.	[112]
			Genus	*Streptococcus*	Increase			
				Enterococcus	Increase			
Allogeneic stem cell transplantation	80 patients	LK (38) LP (21) MM (8) Others (3)	Family	Enterobacteriaceae	Increase	Antibiotic administration during and after: vancomycin, fluoroquinolone, metronidazole, and β-lactam.	The intestinal microbiota may be an important factor in the success or failure in allo-SCT.	[113]
			Genus	*Streptococcus*	Increase			
				*Enterococcus*	Increase			
				*Lactobacillus*	Increase			
Allogeneic stem cell transplantation	31 patients 3 controls	LK (21) LP (4) MM (3) Others (4)	Phylum	Firmicutes	Decrease	Prophylactic antibiotics: trimethoprim/sulfamethoxazole.	*Enterococci* were more prominent and persistent in patients with active acute gastrointestinal graft-versus-host disease.	[114]
			Species	*Enterococcus faecium.*	Increase			
				Enterococcus faecalis	Increase			
Allogeneic stem cell transplantation	209 patients	ALL AML MDS	Family	*Lachnospiraceae*	Decrease	Carbapenems (imipenem or meropenem) and amikacin were administered as first-line antibiotics for patients developing fever during neutropenia.	This study indicated that the intestinal microbiota score could predict survival following allo-SCT.	[115]
				*Ruminococcaceae*	Decrease			
				*Erysipelotrichaceae*	Decrease			
				* Enterobacteriaceae *	Increase			
Allogeneic stem cell transplant	96 patients	AML ALL LP MM	Family	Lachnospiraceae	Decrease	Antibiotics: piperacillin–tazobactam, ticarcillin, meropenem, clindamycin, metronidazole, and intravenous vancomycin.	This study confirms that allo-HSCT-related treatments dramatically alter composition of microbiota, with a marked decrease in its richness and strict anaerobic components.	[94]
				Ruminococacceae	Decrease			
				Enterococcaceae	Increase			
				Staphylococcaceae	Increase			
			Genus	* Streptococcus *	Increase			
Allogeneic stem cell transplantation	100 patients	ALL AML NHL	Genus	* Bacteroides *	Decrease	Antibiotic: vancomycin, fluoroquinolone, and β-lactam.	This study indicates a loss of bacterial diversity after allogeneic HSCT. Gut flora diversity in the engraftment period was a predictor of increased survival.	[116]
				* Enterococcus *	Increase			
				* Klebsiella *	Increase			
				* Escherichia *	Increase			
Autologous stem cell transplantation	30 patients	MM	Family	Lachnospiraceae	Decrease	Six subjects received only levofloxacin as antimicrobial prophylaxis during transplantation.	They suggest that a large loss of bacterial diversity in the immediate post-transplant period is largely due to intravenous antibiotic exposure.	[117]
			Genus	*Ruminococcus*	Decrease			
				*Faecalibacterium*	Decrease			
				*Blautia*	Decrease			
Autologous stem cell transplatation	1362 patients 34 controls	LK LP MM	Genus	* Enterococcus *	Increase	Antibiotic agents: piperacillin–tazobactam and meropenem.	Patterns of microbiota disruption during allogeneic cell transplantation were characterized by loss of diversity and domination by single taxa.	[92]
				* Klebsiella *	Increase			
				* Escherichia *	Increase			
				* Staphylococcus *	Increase			
				* Streptococcus *	Increase			
Stem cell transplantation	11 patients - 8 allogeneic - 3 autologous	AML (8) MM (3)	Phylum: family and genus	*Auto-SCT:* Proteobacteria (*Klebsiella*, *Proteus*, *Acinetobacter*, *Haemophilus*, *Pseudomonas*, Enterobacteriaceae)	Increase	Antibiotic prophylaxis was administered only to allogeneic SCT recipients.	The conditioning regimen promotes changes in the microbiome, which are different between auto- and allo-SCT. Moreover, patients who developed GvHD harbored more Firmicutes and Proteobacteria and less Bacteroidetes than individuals without this complication.	[118]
				* Auto-SCT: * Bacteroidetes (*Bacteroides*, *Saprospirae*, *Prevotella*).	Decrease			
				* Allo-SCT: * Bacteroidetes	Increase			
				* Allo-SCT: * Firmicutes (*Bacilli*, *Lactobacilli*, *Clostridium*, *Enterococci*, *Streptococci*).	Decrease			
Stem cell transplantation	24 patients - 16 allogeneic - 8 autologous 10 controls	LK LP MM	Phylum	Bacteroidetes	Increase	Only two patients received levofloxacin and doripenem because of febrile neutropenia.	They found that the 2-year mortality rate was as high as 66.7% in the cases with the vulnerable microbiotas, whereas the patients who retained the original microbiota composition were alive for at least 2 years after HSCT.	[93]
				Firmicutes	Increase			
				Proteobacteria	Increase			
Stem cell transplant	10 patients	AML ALL	Genus	* Faecalibacterium *	Decrease	All patients received antibiotics for fever and neutropenia during the study period.	These results indicate that the structure and temporal dynamics of the gut microbial ecosystem can be a relevant factor for the success of SCT.	[119]
				* Ruminococcus *	Decrease			
				* Enterococcus *	Increase			
Anti-CD19 CAR T-cell therapy	228 patients 30 controls	NHL ALL	Genus	* Enterococcus *	Increase	Antibiotic: trimethoprim-sulfamethoxazole, intravenous vancomycin, piperacillin–tazobactam, levofloxacin, cefepime, ciprofloxacin, and meropenem.	Higher abundance of *Ruminococcus* was associated with response to CD19 CAR T-cell therapy.	[92]

## Data Availability

No new data were created or analyzed in this study. Data sharing is not applicable to this article.

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
