# Peer review of "Gut Microbiota Disruption in Hematologic Cancer Therapy: Molecular Insights and Implications for Treatment Efficacy"

_ijms, 2024, doi:10.3390/ijms251910255_

Round 1

Reviewer 1 Report

Comments and Suggestions for Authors

The review article by Guevara-Ramírez et al nicely summarizes the current state of anticancer therapy and influence on Gut Microbiota. However, certain sections could benefit from additional clarity regarding the focus on gut bacteria. It is sometimes unclear whether the discussion is centered on the impact of anticancer treatments on gut bacteria or on how gut bacteria might enhance cancer treatment.

Here are some suggestions for the improvement-

1.    Title: Since the review covers how treatments such as chemotherapy and stem cell transplantation can disrupt the gut microbiota which could further impact treatment outcomes and increasing infection risk. Also, if the review is more directed towards bacteria-mediated cancer therapy so the title of the review article should be refined to better illustrates the role of gut microbiota in cancer treatment.  

2.     Review still lacks how gut microbiota-based therapy could be advantageous or should be taken care in patients along with other anticancer therapy. Discuss diet-based interventions, fecal microbiota transplantation and successful outcomes in mouse and human studies in HM.

3.    Provide a more detailed figure on interactions between the gut microbiota and influence on blood cells, how tumor microenvironment is affected by microbiome-secreted metabolites during drug treatments. Discuss drug-microbiome interrelationship in the figure. It should be more detailed and should convey the central narrative of the review article.

4.    Section Routes of Administration and Metabolism is very weak and less informative. The section does not discuss drug-microbiome interrelationship. Either the section can be combined with Drug-Microbiota Interactions or should provide more clear understanding with more research background on route of drug administration and host metabolism influence microbial population.

5.    In the section, Microbial Enzymes and Metabolic Functions please discuss the drugs being used for cancer treatments especially for HM and are there any evidence how microbes affect the drug metabolism. Also, its not clear whether the healthy microbes interfere with drugs or due to drugs the change in microbial composition modulates drug availability. Or combine Microbial Metabolites and their Influence on Drug Metabolism section with this part to give a more comprehensive readout.

6.    What is the relevance of the section Interactions between the Intestinal Microbiota and the Host Immune System? This can be explained in the introduction and discussed in the figure suggested in comment 3.

7.    Why is this discussed separately Metabolism and Enzymatic Degradation? Since it shows repetition and can be combined easily with previous metabolism sections.

8.    It would be nice to include how future chemotherapy agents should be developed, and how microbes and  their derivatives could solve the purpose in the discussion.

Comments on the Quality of English Language

Overall, the English and grammar seem correct. 

Author Response

Reviewer 1

The review article by Guevara-Ramírez et al nicely summarizes the current state of anticancer therapy and influence on Gut Microbiota. However, certain sections could benefit from additional clarity regarding the focus on gut bacteria. It is sometimes unclear whether the discussion is centered on the impact of anticancer treatments on gut bacteria or on how gut bacteria might enhance cancer treatment.

Thank you for your comments and suggestions. All authors appreciate your time and effort.  We have worked to clarify the sections that address both the impact of treatments on the gut microbiota and the potential of gut bacteria to improve cancer treatments. Relevant sections have been revised to ensure that the approach is clear and consistent.

Here are some suggestions for the improvement-

  1. Title: Since the review covers how treatments such as chemotherapy and stem cell transplantation can disrupt the gut microbiota which could further impact treatment outcomes and increasing infection risk. Also, if the review is more directed towards bacteria-mediated cancer therapy so the title of the review article should be refined to better illustrates the role of gut microbiota in cancer treatment.  

Thank you for your comment. We have changed the title to: “Gut Microbiota Disruption in Hematologic Cancer Therapy: Molecular Insights and Implications for Treatment Efficacy”. 

  1. Review still lacks how gut microbiota-based therapy could be advantageous or should be taken care in patients along with other anticancer therapy. Discuss diet-based interventions, fecal microbiota transplantation and successful outcomes in mouse and human studies in HM.

Thank you for your comment. We have expanded this section to include studies on dietary interventions and fecal microbiota transplantation in mice and humans in hematological malignancies. This aspect has been specially developed in the discussion section.

  1. Provide a more detailed figure on interactions between the gut microbiota and influence on blood cells, howtumor microenvironment is affected by microbiome-secreted metabolites during drug treatments. Discuss drug-microbiome interrelationship in the figure. It should be more detailed and should convey the central narrative of the review article.

Thank you for your suggestion. We have created a detailed figure depicting the relationship between drugs and the microbiome, aligning with the central narrative of the article. 

  1. Section Routes of Administration and Metabolism is very weak and less informative. The section does not discuss drug-microbiome interrelationship. Either the section can be combined with Drug-Microbiota Interactions or should provide more clear understanding with more research background on route of drug administration and host metabolism influence microbial population.

Thank you for your comment. We have combined the above sections to improve the flow and provide a more detailed explanation.

  1. In the section, Microbial Enzymes and Metabolic Functions please discuss the drugs being used for cancer treatments especially for HM and are there any evidence how microbes affect the drug metabolism. Also, its not clear whether the healthy microbes interfere with drugs or due to drugs the change in microbial composition modulates drug availability. Or combine Microbial Metabolites and their Influence on Drug Metabolismsection with this part to give a more comprehensive readout.

Thank you for your comment. We have combined the sections on the metabolic functions of microbial enzymes and microbial metabolites to provide a more complete picture. In addition, we have added a detailed discussion on drugs used in cancer treatment, particularly in MH, and evidence on how microbes influence their metabolism. 

  1. What is the relevance of the section Interactions between the Intestinal Microbiota and the Host Immune System? This can be explained in the introduction and discussed in the figure suggested in comment 3.

Thank you for your comment. We have revised and reorganized the section, integrating this information into the detailed figure.

  1. Why is this discussed separately Metabolism and Enzymatic Degradation? Since it shows repetition and can be combined easily with previous metabolism sections.

Thank you for your feedback. We have reorganized the sections to avoid redundancies and improve the flow of the article. However, we have decided to keep the title of the specific paragraph on Metabolism and Enzymatic Degradation in the section dedicated to chemotherapy, as another author suggested integrating the TIMER (Translocation, Immunomodulation, Metabolism, Enzymatic degradation, and Reduced diversity) framework to more accurately describe the interaction between chemotherapy and the microbiota.

  1. It would be nice to include how future chemotherapy agents should be developed, and how microbes and  their derivatives could solve the purpose in the discussion.

Thank you for your suggestion. We have included a section in future directions on the development of future chemotherapeutic agents, considering the influence of the microbiota.

Comments on the Quality of English Language: Overall, the English and grammar seem correct. 

Reviewer 2 Report

Comments and Suggestions for Authors

The manuscript submitted by Patricia et al. titled "Cancer Treatment and Gut Microbiota: Molecular Insights and implications for Hematologic Malignancies’’ offers a thorough examination of the deep connection between cancer treatments, the gut microbiota, and their consequences for hematologic malignancies. The authors have effectively summarized the existing knowledge regarding the impact of several medicines, including as chemotherapy, immunotherapy, and hematopoietic stem cell transplantation, on the gut microbiota and their influence on treatment results. The review also emphasizes the potential of harnessing the microbiome as a therapeutic target to enhance the effectiveness of cancer treatment and minimize negative side effects. The article is found to be interesting and can be published in the journal after working out on the major comments and suggestions. I have compiled a thorough critique that specifically addresses its shortcomings, potential vulnerabilities, areas for further investigation, and any linguistic or technical errors.

1.      The manuscript provides a comprehensive overview of the impact of gut microbiota on cancer treatment, but it lacks precision regarding the precise strains or species of bacteria that have the greatest influence on therapeutic outcomes. Conducting in-depth investigations that specifically target individual microbial species could yield more practical and applicable knowledge.

2.      The review primarily emphasizes the dysbiosis caused by chemotherapy, while providing limited information on the effects of other treatments, such as immunotherapy and stem cell transplantation. Incorporating a more equitable evaluation of different treatment approaches would provide a thorough perspective on the subject.

3.      The manuscript briefly mentions the impact of the microbiota on medication metabolism, but it fails to adequately explore how variations in the composition of an individual's microbiota can result in diverse treatment outcomes. Further exploration of the individualized components of microbiota-mediated therapy could be highly beneficial.

4.      The proposed avenues for further research are very wide-ranging and lack precision. Enhanced recommendations, such as pinpointing precise biomarkers or suggesting novel clinical trials, would offer more distinct guidance for future research.

5.      What is the impact of differences in gut microbiota composition on the effectiveness and side effects of chemotherapy in hematologic malignancies?

In addition, including the details about these questions in the article could also benefit the readers and enhance the quality of the article.

6.      Is it possible to identify particular microbial species that can serve as prognostic biomarkers for patient response to various cancer treatments?

7.      What are the enduring consequences of microbiota-targeted therapy, such as probiotics or fecal microbiota transplantation, on the results of cancer treatment and the overall duration of survival?

8.      What is the specific effect of various cancer treatment methods, such as immunotherapy and radiation, on the gut microbiota?

9.      Is it possible to design individualized techniques to modify microbiota in order to reduce adverse effects associated with treatment and enhance the effectiveness of therapy?

10.  Which microbial species have the most impact on the effectiveness of chemotherapy in hematologic cancers?

11.  What strategies can be employed to manipulate gut microbiota in order to optimize the efficacy of immunotherapy in cancer treatment?

12.  The limitations and benefits of incorporating microbiota-based therapies into established cancer therapy procedures are worth exploring.

Comments on the Quality of English Language

NA

Author Response

The manuscript submitted by Patricia et al. titled "Cancer Treatment and Gut Microbiota: Molecular Insights and implications for Hematologic Malignancies’’ offers a thorough examination of the deep connection between cancer treatments, the gut microbiota, and their consequences for hematologic malignancies. The authors have effectively summarized the existing knowledge regarding the impact of several medicines, including as chemotherapy, immunotherapy, and hematopoietic stem cell transplantation, on the gut microbiota and their influence on treatment results. The review also emphasizes the potential of harnessing the microbiome as a therapeutic target to enhance the effectiveness of cancer treatment and minimize negative side effects. The article is found to be interesting and can be published in the journal after working out on the major comments and suggestions. I have compiled a thorough critique that specifically addresses its shortcomings, potential vulnerabilities, areas for further investigation, and any linguistic or technical errors.

We appreciate your thorough feedback and suggestions. We have carefully reviewed the manuscript to address the areas noted and believe that the modifications made will significantly improve the clarity and scope of the article. Below, we respond to each of the specific points raised.

The manuscript provides a comprehensive overview of the impact of gut microbiota on cancer treatment, but it lacks precision regarding the precise strains or species of bacteria that have the greatest influence on therapeutic outcomes. Conducting in-depth investigations that specifically target individual microbial species could yield more practical and applicable knowledge.

  1. The manuscript provides a comprehensive overview of the impact of gut microbiota on cancer treatment, but it lacks precision regarding the precise strains or species of bacteria that have the greatest influence on therapeutic outcomes. Conducting in-depth investigations that specifically target individual microbial species could yield more practical and applicable knowledge.

Thank you for your comment. We have added sections highlighting specific bacterial species that have a significant impact on cancer treatment outcomes. This information provides a more accurate and applicable approach, aligned with your suggestions.

  1. The review primarily emphasizes the dysbiosis caused by chemotherapy, while providing limited information on the effects of other treatments, such as immunotherapy and stem cell transplantation. Incorporating a more equitable evaluation of different treatment approaches would provide a thorough perspective on the subject.

Thank you for highlighting this area of improvement. We have expanded the sections related to the effects of stem cell transplantation on the intestinal microbiota. Regarding immunotherapy, information is more limited; however, we have included a discussion of this treatment in the early sections of the manuscript to provide a more complete context.

  1. The manuscript briefly mentions the impact of the microbiota on medication metabolism, but it fails to adequately explore how variations in the composition of an individual's microbiota can result in diverse treatment outcomes. Further exploration of the individualized components of microbiota-mediated therapy could be highly beneficial.

We agree with your observation. We have revised the discussion to include a more in-depth analysis of this aspect.

  1. The proposed avenues for further research are very wide-ranging and lack precision. Enhanced recommendations, such as pinpointing precise biomarkers or suggesting novel clinical trials, would offer more distinct guidance for future research.

We appreciate this suggestion. We have refined the recommendations for future research and new directions for clinical trials exploring microbiota modulation in combination with standard therapies have also been suggested.

  1. What is the impact of differences in gut microbiota composition on the effectiveness and side effects of chemotherapy in hematologic malignancies?

We have expanded the discussion in the manuscript to address how variations in the gut microbiota may affect both the efficacy of chemotherapy and associated side effects.

In addition, including the details about these questions in the article could also benefit the readers and enhance the quality of the article.

  1. Is it possible to identify particular microbial species that can serve as prognostic biomarkers for patient response to various cancer treatments?

Yes, it is possible. We have added a detailed discussion in the manuscript on microbial species that are being investigated as possible prognostic biomarkers.

  1. What are the enduring consequences of microbiota-targeted therapy, such as probiotics or fecal microbiota transplantation, on the results of cancer treatment and the overall duration of survival?

We have expanded the corresponding section to include an analysis of the long-term effects of microbiota-targeted therapies.

  1. What is the specific effect of various cancer treatment methods, such as immunotherapy and radiation, on the gut microbiota?

We have expanded the manuscript to provide a more detailed discussion of this aspect.

  1. Is it possible to design individualized techniques to modify microbiota in order to reduce adverse effects associated with treatment and enhance the effectiveness of therapy?

Yes, it is possible. Tailoring interventions on the microbiota according to patient-specific factors holds significant therapeutic potential that is under investigation. An example was added in the article concerning the modulation of the microbiota to avoid adverse effects of a drug.

  1. Which microbial species have the most impact on the effectiveness of chemotherapy in hematologic cancers?

We have identified and discussed in the manuscript several microbial species that have a significant impact on the effectiveness of chemotherapy in hematologic cancers.

  1. What strategies can be employed to manipulate gut microbiota in order to optimize the efficacy of immunotherapy in cancer treatment?

This article discusses strategies based on the manipulation of the intestinal microbiota to improve the efficacy of immunotherapy in cancer treatment, including the use of probiotics, prebiotics and fecal microbiota transplantation

  1. The limitations and benefits of incorporating microbiota-based therapies into established cancer therapy procedures are worth exploring.

Thank you for your comments, the limitations and benefits of incorporating microbiota-based therapies were added.

Comments on the Quality of English Language NA

Reviewer 3 Report

Comments and Suggestions for Authors

Hematologic malignancies (HMs), including leukemia, lymphoma, and multiple myeloma, involve the uncontrolled proliferation of abnormal blood cells, posing significant clinical challenges due to their heterogeneity and varied treatment responses. Despite recent advancements in therapies that have improved survival rates, particularly in chronic lymphocytic leukemia and acute lymphoblastic leukemia, treatments like chemotherapy and stem cell transplantation often disrupt gut microbiota, which can negatively impact treatment outcomes and increase infection risks. This review explores the complex, bidirectional interactions between gut microbiota and cancer treatments in patients with HMs. Gut microbiota can influence drug metabolism through mechanisms such as the production of enzymes like bacterial β-glucuronidases, which can alter drug efficacy and toxicity. Moreover, microbial metabolites like short-chain fatty acids can modulate the host immune response, enhancing treatment effectiveness. However, chemotherapy often reduces the diversity of beneficial bacteria, such as *Blautia*, *Prevotella*, and *Faecalibacterium*, while increasing pathogenic bacteria like *Enterococcus* and *Escherichia coli*. These findings highlight the critical need to preserve microbiota diversity during treatment. Future research should focus on personalized microbiome-based therapies, including probiotics, prebiotics, and fecal microbiota transplantation, to improve outcomes and quality of life for patients with hematologic malignancies.

The topic of this review is of relevance for the scientific community and I think worth being published. Yet, the manuscript (at least in its current form) appears preliminary and not really carefully crafted, resembling more a draft than a final version.

The following pertinent reports should be mentioned/discussed:

doi: 10.1016/j.phrs.2023.106891.

doi: 10.3390/nu15173680.

doi: 10.3390/nu15153327.

doi: 10.3390/nu15194253.

doi: 10.3389/fendo.2023.1265152.

doi: 10.1016/j.phrs.2023.106870.

The presentation and critical interpretation of results of previous studies should be improved.

The Authors should incorporate informative elements for Readers in figures and tables (seeking professional assistance is advisable).

Comments on the Quality of English Language

-

Author Response

Thank you for your valuable comments. We have revised the manuscript to integrate the suggested references into the study's approach, incorporating them into the discussion to offer a more complete and updated view of the topic. In addition, we have optimized the presentation and critical interpretation of the results of previous studies, highlighting more clearly the most relevant contributions to our analysis.

Regarding the figures and tables, we have incorporated another figure that summarizes the content presented and the tables represent the summary of the reviewed analyses. We thank you again for your recommendations, which have been of great help in strengthening our work.

Round 2

Reviewer 2 Report

Comments and Suggestions for Authors

The authors had worked out on the comments and i suggest its acceptance.

Author Response

We sincerely appreciate your positive evaluation and comments during the review process. We are pleased to know that the modifications made meet your expectations and that our work has been considered for acceptance.

We are very grateful for your time and valuable contributions, which have significantly improved the quality of our manuscript.

Reviewer 3 Report

Comments and Suggestions for Authors

It is unclear where the suggested reports have been discussed.

Comments on the Quality of English Language

-

Author Response

We sincerely appreciate your valuable comments and suggestions, which we have carefully reviewed. We have incorporated three of the proposed revisions (doi: 10.3390/nu15173680 (181), doi: 10.3390/nu15153327 (184), doi: 10.3390/nu15194253 (183)), particularly those related to microbiota and diet, as we believe they provide important future directions for our research and align closely with the central focus of our study.

The suggested references have been included in lines 1116-1131 of page 62.

“Undoubtedly, the treatment of hematologic cancer has generated a growing interest in the modulation of the intestinal microbiota through probiotics, prebiotics, and fecal microbiota transplantation. However, these therapies present significant limitations and challenges that must be carefully evaluated in their clinical application. While probiotics, prebiotics, and FMT can offer important benefits, especially after oncologic treatment, their use also carries potential risks and side effects. The main limitation is the lack of solid evidence to support their use reliably and effectively. In addition, individual response to these therapies is highly variable; for example, prebiotics rich in carbohydrates can cause bloating and abdominal discomfort in some patients [179, 180]. Moreover, dysbiosis can also be improved through diet. Recent studies suggest that the ketogenic diet may modify the diversity and composition of the gut microbiota, which in turn could influence the efficacy of treatment for conditions such as cancer, epilepsy, and obesity [181, 182]. Another study on acute leukemia suggests that a diet rich in vegetables and fruits modulates microbiota and immune responses [183]. They also suggest that exposure to plant diversity rich in glucan and related microbial communities promotes immune cell maturation and is linked to a lower incidence of childhood acute lymphoblastic leukemia [184].”

Regarding the other three suggestions (doi: 10.1016/j.phrs.2023.106891, doi:10.3389/fendo.2023.1265152, doi: 10.1016/j.phrs.2023.106870), while we acknowledge their scientific relevance, we chose not to incorporate them into this version of the manuscript as they pertain to areas such as colon cancer, hyperandrogenism, and microRNA-based therapies for inflammatory brain disorders. Although these topics are of significant interest, they are not directly related to the primary scope of our article, which focuses on the relationship between the microbiota and hematologic diseases.

Additionally, we have performed a thorough revision of the manuscript's English language to ensure clarity and accuracy.

We remain at your full disposal for any further adjustments or suggestions you may have.

Round 3

Reviewer 3 Report

Comments and Suggestions for Authors

-

Comments on the Quality of English Language

-